



# Experiments with large number of GNSS-RO observations through the ROMEX collaboration in the Met Office NWP system

Neill E. Bowler[1] and Owen Lewis[1]

[1]Met Office, Fitzroy Road, Exeter, EX1 3PB, UK.

**Correspondence:** Neill E. Bowler (neill.bowler@metoffice.gov.uk)

**Abstract.** Over recent years there has been an increase in the number of GNSS-RO observations available for use in numerical weather prediction (NWP). The Radio Occultation Modelling Experiment (ROMEX) was set up to assess the impact of increasing numbers of GNSS-RO observations and to provide evidence for a further increase in the number. Unlike previous studies, ROMEX gathered a large set of real observations to test the impact, rather than use simulated observations.

Tests assimilating the ROMEX observations into the Met Office's NWP system showed negative impacts on forecast quality. This was largely due to a bias in the forecasts of geopotential height in the troposphere. This bias is shown to be due to the impact of GNSS-RO observations in the stratosphere. The forward operator for GNSS-RO has a long "tail" meaning that the NWP system is able to adjust the height of observations in the stratosphere by changing the tropospheric state (i.e. raising or lowering the height of the observation). Thus, the NWP system can adjust to a systematic difference between the model and observations in the stratosphere by creating a bias in the tropospheric state. Therefore, it was necessary to adjust the refractivity operator used in the forward model to reduce this bias in the forecast state. After much experimentation it was decided to alter the refractivity coefficients in the operator by 0.1% and 3.5% for $k_1$ and $k_2$, respectively. When using this adjusted operator it was possible to demonstrate the beneficial effect of assimilating the ROMEX observations.

Additional modifications were also applied to the processing of GNSS-RO observations, including vertical smoothing of the observed profiles, and a bias correction at high altitudes to correct for errors within the NWP model. With this modified operator various experiments were conducted to assess impact of increases to the total number of observations. It was shown that the addition of the extra ROMEX observations provided a substantial improvement in the forecast quality. This is particularly true in the southern-hemisphere extra-tropics where the largest benefits were seen for the additional data. The benefit seen for a given number of additional observations varied substantially with the region being considered and the data against which the verification was being performed. Overall the largest forecast improvements were seen when assimilating 20,000 occultations per day, although this may be connected to the quality of the observations being excluded when reducing the ROMEX dataset to this number.



# 1 Introduction

## 1.1 The use of GNSS-RO observations in NWP

Numerical weather prediction (NWP) has brought great benefit to society over many years. NWP forecasts have been gradually improving in quality over time Bauer et al. (2015). This is due to improved numerical models, increased numbers of observations and improved methods for assimilating those observations.

Radio occultation (RO) observations from global navigation satellite systems (GNSSs) provide one source of observations to the global observing system. These measurements use the time delay in receiving signals from GNSS satellites at a low-earth orbiting (LEO) satellite, caused by refraction of these signals in the earth's atmosphere. As the LEO orbits the earth the GNSS satellite will appear to rise above or set below the earth's horizon, depending on the relative motion of the satellites. Thus a sequence of observations of the amount of refraction provides a profile through the earth's atmosphere — each such profile is referred to as an occultation. For more information, see for instance Kursinski et al. (1997); Anthes (2011).

Over recent years the number of GNSS-RO observations has been steadily increasing. This has been largely driven by the launch of the COSMIC-2 constellation Schreiner et al. (2020), Sentinel-6A and the purchase of observations from private companies such as Spire Bowler (2020a) and PlanetIQ Mo et al. (2024).

The Coordination Group for Meteorological Satellites (CGMS) of the World Meteorological Organisation (WMO) provides recommendations on the number of observations that should be made each day by the various observation platforms. Whilst the CGMS is not able to mandate the number of observations to be made, it does provide guidance to the WMO members on the number of observations that should be made. The CGMS high-level priority plan provides the following recommendation Coordination Group for Meteorological Satellites (2024b):

Advance the atmospheric radio occultation constellation, with the long-term goal of providing 20000 occultations per day with uniform spatial and local time coverage on a sustained basis.

The CGMS also provides a baseline document which further states the target Coordination Group for Meteorological Satellites (2024a):

Minimum 6000 occultations from low inclination orbits ($<30°$) distributed geographically and temporally in local time, 1000 occultation from other drifting orbits, and 7600 occultations from sun-synchronous orbits. Electron density profiles up to 500 km.

Whilst the number of observations has increased in recent years, the current number of occultations available daily (approximately 13,500) is still below the figure from the baseline document (14,600 occultations per day) and far below the target of 20,000 occultations per day in the high-level priority plan.

## 1.2 ROMEX

The Radio Occultation Modelling Experiment (ROMEX, Anthes et al. (2024)) was developed as an approach to provide evidence of the impact of increased numbers of GNSS-RO observations on NWP. Previous experiments have considered the



impact of simulated observations. However, it was noted that a large number of GNSS-RO observations were available during 2022, largely from commercial providers. With the acquisition of these observations it would be possible to test the impact of

increasing the number of observations available without the limitations caused by the use of simulated observations.

ROMEX brought together a large number of organisations from around the world. This included a variety of of data providers from Europe, the USA and China, including commercial companies from within those regions. These organisations made their observations available to the project. Most of the data used in ROMEX was processed to bending angles by either the European Organisation for the Exploitation of Meteorological Satellites (EUMETSAT) or the University Corporation for Atmospheric

Research (UCAR), which helped to minimise differences or problems in the data. These data were provided to NWP centres within the ROMEX consortium in order to assess the impact of the data on NWP. It was requested that all NWP centres run at least the control experiment (consisting of data from the Metop, COSMIC-2, KOMPSAT-5, PAZ, TerraSAR-X, TanDEM-X and Sentinel-6A satellites) and an experiment with the entire dataset. Additional experiments were suggested, but these were considered optional.

## 70  1.3  Previous experiments

The expected impact of increasing the number of GNSS-RO observations on NWP was studied by Harnisch et al. (2013) using an ensemble of data assimilations (EDA). In that study synthetic observations were created, and the impact of the observations were estimated by examining the change in the spread of the ensemble. They simulated 128,000 occultations per day and the effect of using certain fractions of these observations was considered. They found that 16,000 occultations per day gave

approximately half the reduction in the ensemble spread of using the full set of observations. Therefore they recommended that 16,000–20,000 globally distributed occultations per day be considered as a minimum requirement for the global observing system.

Prive et al. (2022) conducted an obsevation system simulation study (OSSE) to assess the impact of increasing the number of GNSS-RO observations. An OSSE uses synthetic observations, like an EDA study. However, the simulated observations are

derived using a "nature run" from a model which is independent of the model used to produce the forecasts. Thus the effects of differences between the models will affect the use of the observations. For this reason, we might expect that the results from an OSSE would be more reliable than those from an EDA. They simulated up to 100,000 occultations per day and found that there was no saturation in the benefit to the forecast from assimilating additional data. However, they noted that the rate of improvement diminished after 50,000 occultations per day was reached. They speculated that saturation of benefit might be

reached at around 150,000 occultations per day in their system. Their experiment found forecast degradation in certain regions related to the addition of GNSS-RO observations — this was believed to be due to sub-optimal specification of the observation uncertainties for the GNSS-RO data.



## 2 Initial experiments

All the experiments documented in this report were run using the low-resolution version of the Met Office's NWP trial work-
flow. The resolution of the NWP forecast is described as N320, meaning that it has 640 by 480 grid points, corresponding to a
resolution of around 40 km in the mid-latitudes. The model also uses 70 levels in the vertical, stretching from 20 metres above
the surface to 80 km altitude. The forecasts are for the atmosphere only, and take a prescribed sea-surface temperature from the
OSTIA data assimilation system Fiedler et al. (2019). The data assimilation system is a hybrid 4D-Var system, meaning that
a portion of the background-error covariances are derived from the operational ensemble. The system is run in "uncoupled"
mode, meaning that an ensemble forecast is not run as part of the experiments, but the ensemble information is taken from the
archive of the operational system.

For the purposes of running the ROMEX experiments, none of the operationally-available GNSS-RO observations were
assimilated. Instead all the GNSS-RO observations are taken from the ROMEX dataset, either for the control set of observations
or for the experiment with additional observations. The initial experiment was to compare the control experiment with the
experiment with all the ROMEX observations.

### 2.1 Initial results

Headline results from the initial experiments are shown in Figure 1. Whilst there is an improvement in the forecast for some
quantities, such as tropical winds and temperatures at certain heights, there is a large degradation in the extra-tropical forecasts
of geopotential height. This is most apparent when considering verification against ECMWF analyses at short lead times. The
cause of this degradation is due to a large shift in the bias of the geopotential height forecasts — the geopotential heights in
the troposphere are much reduced when assimilating the additional observations. Whilst it is hard to be certain about the true
value of the average geopotential height of a given pressure level, the shift in the bias is large enough to be confident that it is
detrimental change.

### 2.2 Investigating the bias by adjusting observations

Since the largest detrimental impacts were seen in geopotential heights which are related to the integral of atmospheric tem-
perature from the surface, it was thought that the sources of the bias would be related to observations in the lower troposphere.
Therefore, an experiment was run which adjusted the GNSS-RO observations in this region. Figure 2 shows the mean difference
between the observations and the model background of the observations, normalised by their average, that is

$$\mu = \frac{2}{N} \sum_{i=1}^{N} \frac{O_i - B_i}{O_i + B_i} \tag{1}$$

where $O_i$ and $B_i$ are the observation and model background values for observation $i$, respectively, and $N$ is the number of
occultations. Each profile of observations is interpolated to a 200 m grid, thus the number of observations is approximately
constant in height.





**Figure 1.** Scorecard showing the change in root-mean-square forecast error for the experiment with all ROMEX observations, compared to the control. The experiments are verified against ECMWF operational analyses (left) and observations (right). The area of the triangles is proportional to the fractional change in RMSE. Green (purple-blue) triangles indicate better (worse) performance for the experiment. The y-axis denotes the forecast variable being considered. NH, TR and SH denote the northern hemisphere > 20 degrees, tropics and southern hemisphere < −20 degrees, respectively. The letter following the underscore indicates the weather parameter, with W, T and Z signifying vector wind field, temperature and geopotential height, respectively. The numbers after this indicate the height of the observation in hPa, except for 2m and 10m that are heights above the surface. The x-axis shows the forecast lead time in hours.



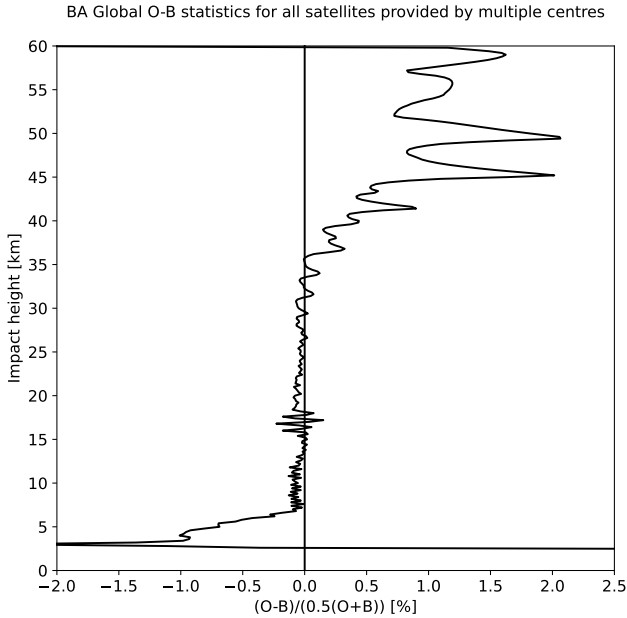

**Figure 2.** Mean difference between the observations and the model background of the observations, normalised by their average. The data are averaged over all the ROMEX observations.

In the lower troposphere the mean values in Figure 2 are substantially below zero, indicating that the observations are consistently smaller than the model background. This is a region where data processing choices can affect the bias. For instance, 120 the length of the data record into the earth's shadow can affect lower-tropospheric biases Sokolovskiy et al. (2010). If we adjust the observations in this region to be larger, then the bias in O-B in this region will be reduced. An experiment was run where the observations were adjusted by a factor, linearly increasing from 1 (no adjustment) at 7 km impact height, to 1.025 at 0 km impact height (noting that this is below the earth's surface). This increase is designed to approximately fit the O-B bias seen in Figure 2. The results of this experiment are shown in Figure 3. Whilst the overall results have improved, there are still 125 substantial degradations in many forecast quantities relative to the control. In particular, the geopotential height forecasts have improved, but are still substantially degraded compared to the control, especially when verified against ECMWF analyses.

Whilst the geopotential height at 500 hPa represents an integral of the atmospheric state from the surface up to 500 hPa, there is some evidence that we should be considering observations which are located above this level. Eyre (1994) noted that the forward operator for bending angle observations contains sensitivity to the atmospheric state well below the height of the 130 tangent point of the ray. An adjustment to the atmospheric state below the observation will alter the modelled height of the observation, thus adjusting the model forecast of the observation. This has been referred to as a hydrostatic tail for the operator Bauer et al. (2014). Given this behaviour, one might ask whether the observations at higher altitudes can affect the behaviour of the forecasts of 500 hPa geopotential height. Looking again at 2, we notice that there is a small negative bias to the O-B







**Figure 3.** Scorecard showing the change in root-mean-square forecast error for the experiment with all ROMEX observations but with a bias adjustment to observations in the lower troposphere, compared to the control. Figure format as in Figure 1.



statistics between approximately 7 and 30 km impact height. This region is known colloquially as the "golden region", since
it is where the observations are most accurate and are given the highest weight in data assimilation systems. Therefore, it is
possible that very small biases in this region can play a role in forecast quality.

   To test this hypothesis, we ran a further experiment where the observations were increased by 0.05% throughout the entire
profile, since 0.05% is approximately the difference in the O-B statistics between 7 and 30 km impact height. The results
of this experiment are shown in Figure 4. This experiment performs much better than the initial experiment, especially for
tropospheric geopotential heights. However, the overall performance is still negative for many forecast variables, and therefore
we run a further experiment where all observations are increased by 0.1%. The results of this experiment are shown in Figure
5. These results show that by bias correcting the GNSS-RO observations by 0.1% we achieve a situation where the addition of
the ROMEX observations is now a positive change overall, rather than a negative one. Certain results are still negative, such as
the 50 hPa geopotential height forecasts measured against ECMWF analyses, and some of the medium-range forecasts when
measured against radiosondes. Hence, this bias correction of the observations cannot be regarded as a complete solution.

   It was previously stated that much of the degradation in the forecast quality was due to a change in the bias of geopotential
height forecasts in the troposphere. Figure 6 shows the change in the bias of the 500 hPa geopotential height forecasts for the
control and various experiments which assimilate all the ROMEX observations. This shows a large reduction in the forecasts
of 500 hPa geopotential height with the initial experiments assimilating all the ROMEX experiments. The experiment which
adjusts the observations by 0.05% approximately halves this reduction, and the experiment adjusting by 0.1% eliminates it
entirely. This seems to be the main reason that the adjusted experiments perform better than the initial experiment — they are
able to remove the large negative bias in the geopotential height forecasts.

## 2.3    Demonstrating dependence using 1D-Var

To further understand how this bias arises, we use a 1-dimensional variational (1D-Var) assimilation to examine the analysis
increments. A 1D-Var assimilation is run as part of the quality control of observations at the Met Office. If the minimisation
does not converge within 20 iterations, then the entire profile of observations is rejected Bowler (2020b). The 1D-Var process
is not identical to treatment of observations within the 4D-Var minimisation used to initialise the forecasts, since no account
is made for the drift of the tangent point, and the same 1D background-error covariance matrix is used for all observations.
Nonetheless, analysis increments from this system can provide useful insights.

To demonstrate the behaviour of the ROMEX experiments, we take observations from the second cycle of the experimental
period – centred around 0600 UTC on 1st September 2022. The model background state is taken from the operational suite on
that day, and all the ROMEX observations between 0300 and 0900 UTC are used in the processing. To show how observations
at different heights affect the increment produced by the 1D-Var analysis, the observations were perturbed by multiplying them
by the factor 1.001 in various height ranges. The observations were only perturbed if their impact height was within 2.5 km of
a given value, where the value was either 5, 10, 15 or 20 km. These perturbed analysis increments were then compared against
an analysis without any perturbation.



**Figure 4.** Scorecard showing the change in root-mean-square forecast error for the experiment with all ROMEX observations but with a bias adjustment of 0.05% to all observations, compared to the control. Figure format as in Figure 1.







**Figure 5.** Scorecard showing the change in root-mean-square forecast error for the experiment with all ROMEX observations but with a bias adjustment of 0.1% to all observations, compared to the control. Figure format as in Figure 1.





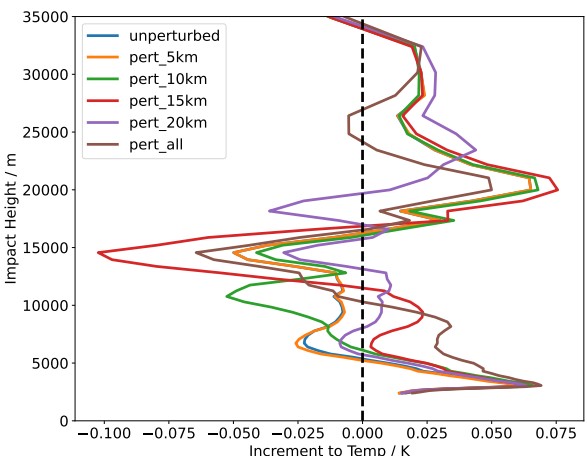

**Figure 6.** Bias for 500 hPa geopotential height forecasts, measured against ECMWF analyses. The control experiment (black) is compared to the experiment with all ROMEX observations (red), the experiment with a bias correction of 0.05% (orange) and the experiment with a bias correction of 0.1% (blue).

Figure 7 shows the analysis increment from the perturbed and unperturbed experiments for pressure, temperature and specific humidity, averaged over all the observations. Since it is a global average, the increment is rather small and potentially changed substantially by the perturbations. Notably, the average increment to pressure is negative below 23 km impact height whereas the temperature increment changes sign at various points in the profile. The average increment to specific humidity is negative, and very similar for both the perturbed and unperturbed simulations.

It is interesting to consider the difference between the analysis increments from the perturbed and unperturbed systems. This is shown in Figure 8 for temperature, pressure and specific humidity. For temperature, the main change in the increment is a reduction in the temperature at the region where the observations have been perturbed, and an increase at other levels. For pressure, the maximum change is at the bottom of the perturbed region, with an increase in the pressure both within and below





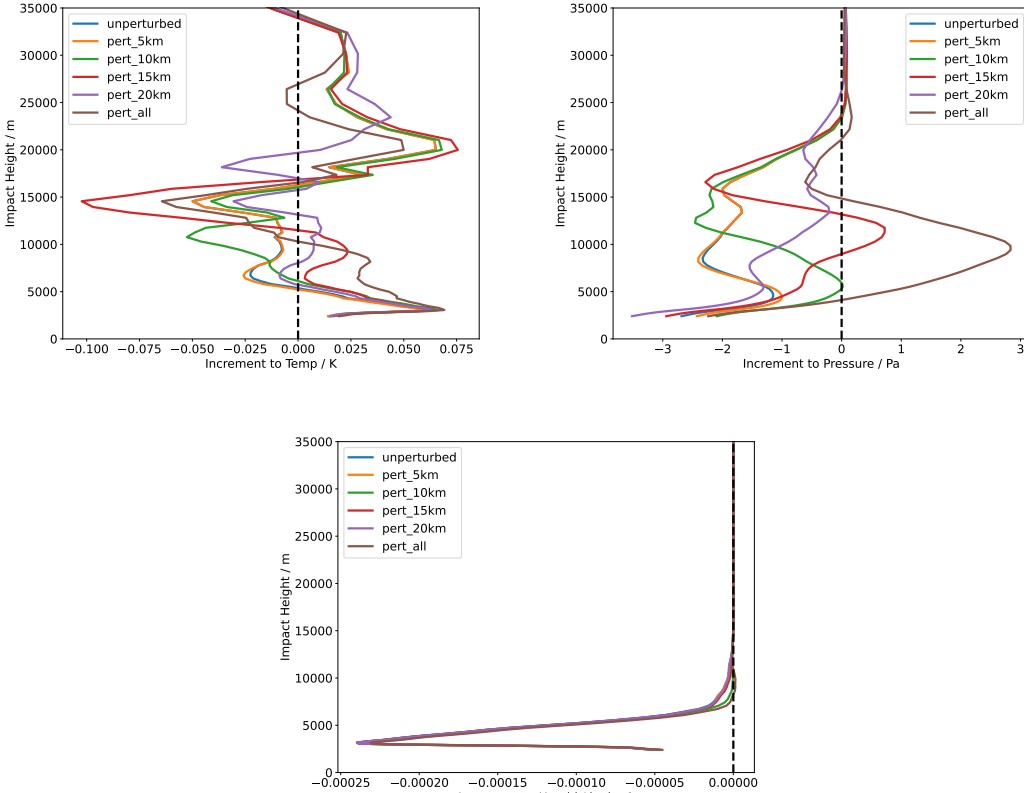

**Figure 7.** Average analysis increments to temperature (top-left), pressure (top-right) and specific humidity (bottom-left) from the 1D-Var system. Shown are increments from the unperturbed system (blue) and those with perturbations centred on a specific region.

the perturbed region. The perturbations to specific humidity serve to increase it, with the largest changes for those perturbations which occur in the troposphere.

The above results indicate that the change in tropospheric geopotential height analyses are likely due to a systematic reduction in the atmospheric pressure for the addition of GNSS-RO observations. As was noted by Eyre (1994) the forward operator for bending angle observations means that the analysis increments can extend well below the region of the observations themselves

## 2.4 Adjustments to the refractivity operator

Following the experiments adjusting the observations by a constant factor, it was pointed out (Katrin Lonitz, personal communication) that a very similar effect could be achieved by adjusting the refractivity coefficients in the observation operator. To calculate the refractivity from a model forecast, the Met Office uses the two-term formula of Smith and Weintraub (1953)

$$N = \frac{k_1 p}{T} + \frac{k_2 e}{T^2} \tag{2}$$



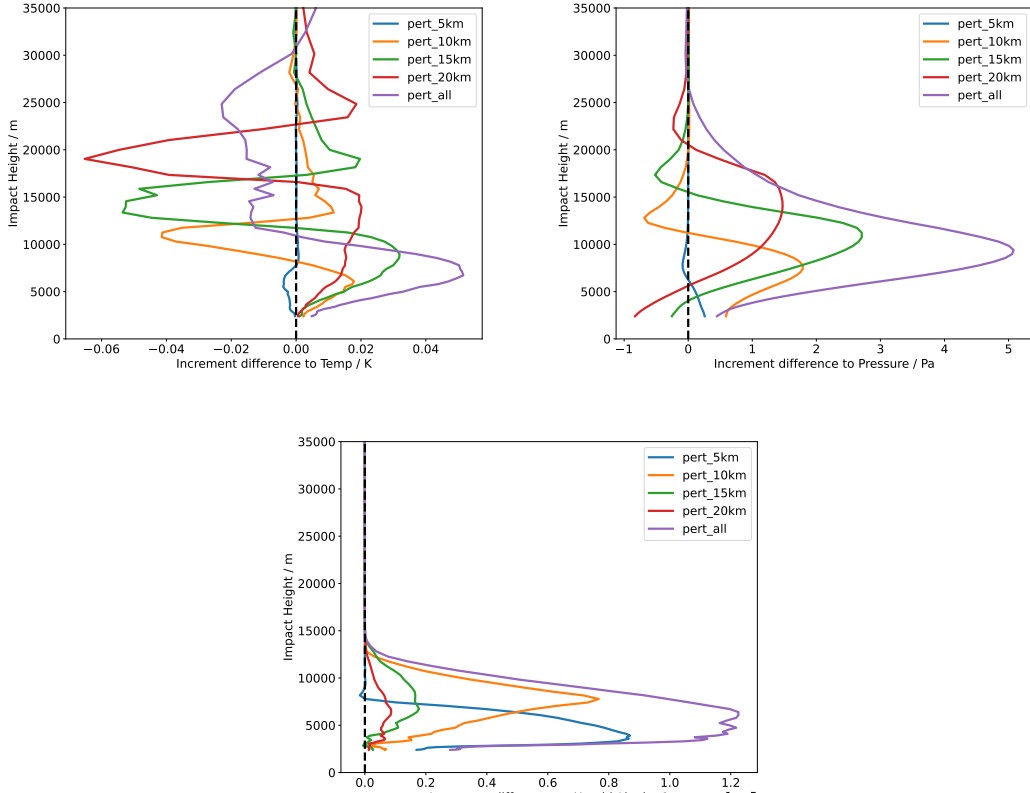

**Figure 8.** Difference in the analysis increments between the perturbed and unperturbed assimilations. Differences are shown for pressure (top-left), temperature (top-right) and specific humidity (bottom-left).

where $p$ is the total atmospheric pressure (including both the dry gases and water vapour, in Pa), $T$ is the temperature in Kelvin, $e$ is the partial pressure of water vapour, also in Pa, $k_1$ and $k_2$ are constants derived from laboratory measurements and are $k_1 = 0.776$ and $k_2 = 3.73 \times 10^3$. The constants used in this equation are derived from rather old experiments, and more recent formulations of the refractivity may be preferred Healy (2011); Aparicio and Laroche (2011). Healy (2011) quotes an estimated error of around 0.1% in the formula for refractivity. Since it is derived from first principles, Aparicio and Laroche (2011) give an estimated error of 0.01% in their formula. Therefore, assuming changes of 0.1% in the formula of Smith and Weintraub (1953) is reasonable.

To run an experiment equivalent to the ones above, $k_1$ was reduced by 0.1%, and the results are shown in Figure 9. There is evidently strong similarity between the results of this experiment and the ones shown in Figure 5, although not exactly equivalent.

Considering Figure 2 once more, one will note that in addition to the very small bias between 7 and 30 km impact height, there are also large biases in the lower troposphere, as noted previously. Therefore, we consider combining changes to both $k_1$ and $k_2$ in the refractivity operator. In order to achieve an approximately unbiased average value of the O-B statistics we reduce



**Figure 9.** Scorecard showing the change in root-mean-square forecast error for the experiment with all ROMEX observations but with a reduction by 0.1% of $k_1$ in the refractivity operator, compared to the control. Figure format as in Figure 1.



$k_1$ to 0.775612 (a 0.05% reduction) and $k_2$ to 3600 (a 3.5% reduction). Whilst we cannot offer any theoretical justification for either of these changes, they are chosen to approximately match the O-B statistics. The results of this experiment are shown in Figure 10. Aside from high altitude geopotential heights against ECMWF analyses, this experiment performs well, with many more positive results than negative ones. The results for geopotential heights in the troposphere are now generally positive, indicating that the former bias issues have been effectively addressed.

## 2.5   Data assimilation statistics

Another important measurement of the quality of a given change to the data assimilation system is the fit of other observations to the model background. We measure this using the standard deviation of the difference between the observations and either the model background or the analysis. The ratio of this standard deviation, relative to the same quantity for the control is plotted for each experiment. If this ratio is less than one then it indicates that the experiment has a better fit to the independent observations than the control. Figure 11 shows this ratio for various observation groups using the background forecast for the experiment which reduced $k_1$ by 0.05% and $k_2$ by 3.5%. For all three instruments shown, the experiment is closer to the observations for many of the channels. However, for each of the instruments there are some channels where the ratio is greater than one, indicating that the fit to the observations is worse than the control. Cross-track infrared sounder (CrIS) channels with numbers below 25, and Infrared Atmospheric Sounding Interferometer (IASI) channels with numbers below 70 are typically temperature sounding channels which peak in the stratosphere. Similarly, Advanced Microwave Sounding Unit (AMSU-A) channel 8 peaks in the lower stratosphere. Therefore, all those channels for which the addition of GNSS-RO observations has a negative impact are in the stratosphere.

Figure 12 shows the same statistics as Figure 11, but for the analysis rather than the background. We see that the fit to the independent observations is degraded for almost all channels when using the analysis. In a perfect system you might expect that adding extra GNSS-RO observations would improve the representation of truth in the analysis leading to improvements of fits to other observations. This is often not the behaviour observed due to differences between observation types including their resolution, coverage, bias etc, which means that you often find adding much more of one dataset pulls the analysis away from some others.  The overall skill of the analysis might still be improved and give rise to the more widespread (although not universal) improvements in short-range forecast fits.

## 2.6   Standard deviation of forecast error

The verification results presented above are based on the root-mean-square error of the forecast. It is common instead to consider the standard deviation of the forecast error, as this is unaffected by forecast biases. Typically, the root-mean-square error is calculated using the following formula

$$RMSE = \sqrt{\frac{1}{N_t}\sum_{i=1}^{N_t}\frac{1}{N_o}\sum_{i=1}^{N_o}(f_{i,t}-o_{i,t})^2} \qquad (3)$$



**Figure 10.** Scorecard showing the change in root-mean-square forecast error for the experiment with all ROMEX observations but with a reduction by 0.05% of $k_1$ and a reduction of 3.5% of $k_2$ in the refractivity operator, compared to the control. Figure format as in Figure 1.





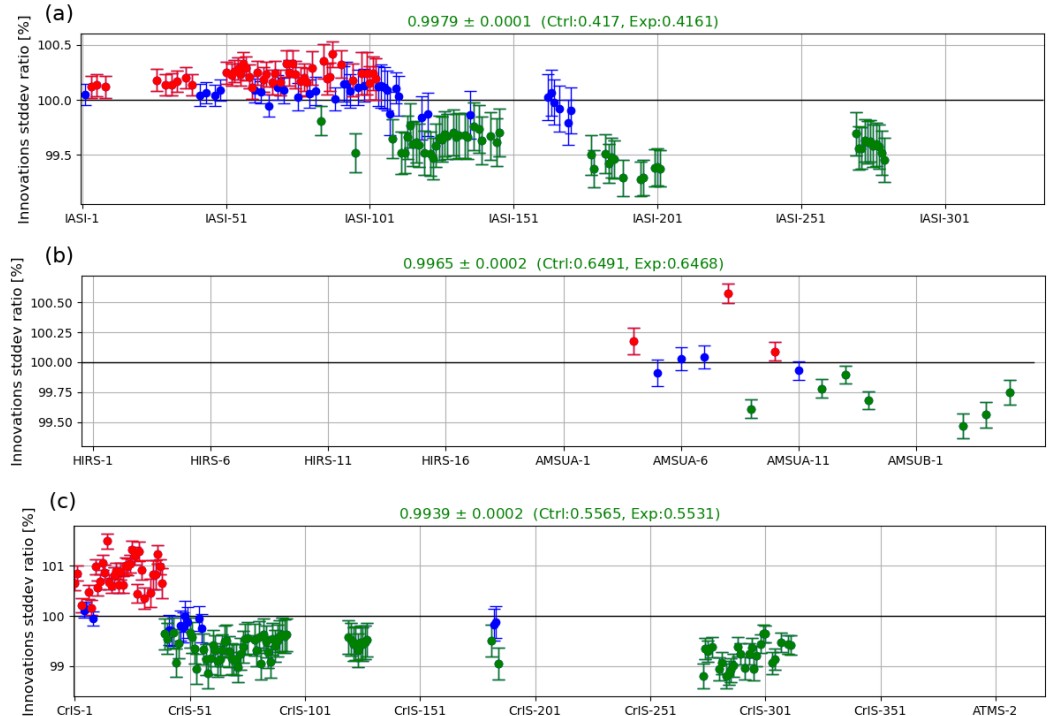

**Figure 11.** Ratio of the standard deviation of the difference between the observations and model background for the experiment with all ROMEX observations but with a reduction by 0.05% of $k_1$ and a reduction of 3.5% of $k_2$ in the refractivity operator, compared to the control. Statistics are calculated against (a) IASI observations on Metop-B, (b) ATOVS observations on Metop-C and (c) CrIS observations on NOAA-20. The x-axis shows the Met-Office channel number, which is based on a subset of the full channel set that the Met Office receives for CrIS and IASI.

where $o_{i,t}$ is the observation at time $t$ and location $i$, $f_{i,t}$ is the forecast of this observation, and $N_t$ and $N_o$ are the number of times and locations, respectively. This means that the mean-square error is calculated for each observation time, and then this is averaged over all times. In a similar way, we can calculate the forecast bias via

$$\text{Bias} = \frac{1}{N_t} \sum_{i=1}^{N_t} \frac{1}{N_o} \sum_{i=1}^{N_o} (f_{i,t} - o_{i,t}). \tag{4}$$

The calculation of the standard deviation of the forecast error is based on these two quantities

$$\sigma = \sqrt{RMSE^2 - \text{Bias}^2} \tag{5}$$

where $\sigma$ is the standard deviation of the forecast error.

Experiments at ECMWF and DWD (Katrin Lonitz, Harald Anlauf, personal communications) demonstrated that the addition of ROMEX observations, without any changes to the operator to account for the bias, resulted in reductions in the standard deviation of the forecast error. Figure 13 shows a scorecard of the standard deviation of the forecast error for the experiment





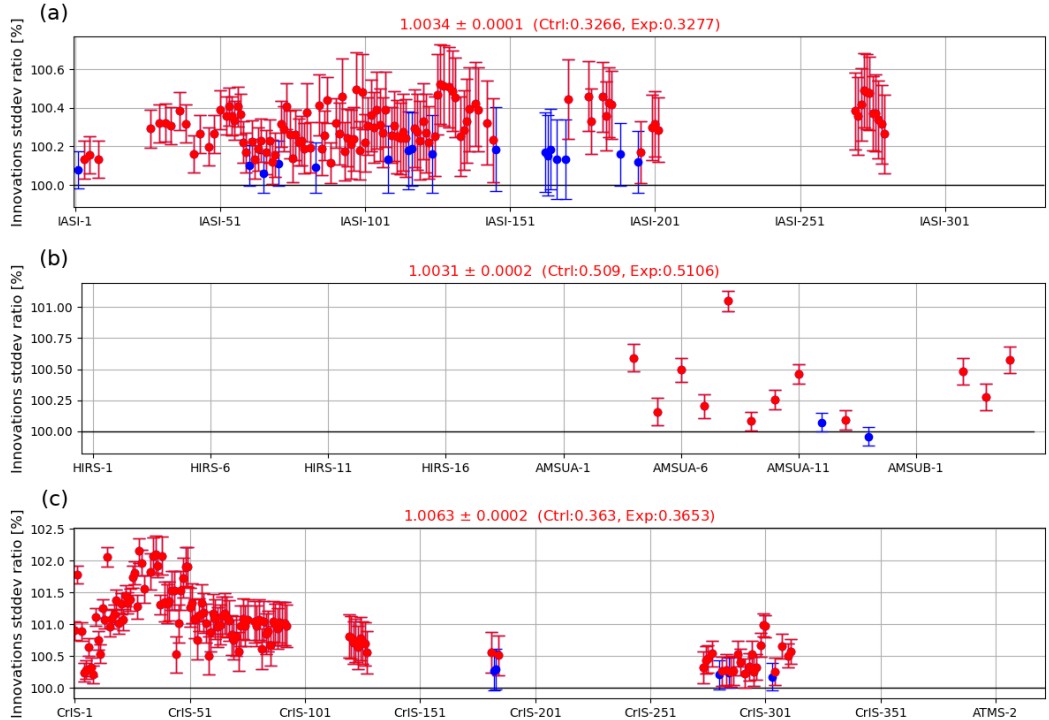

**Figure 12.** Ratio of the standard deviation of the difference between the observations and model analysis for the experiment with all ROMEX observations but with a reduction by 0.05% of $k_1$ and a reduction of 3.5% of $k_2$ in the refractivity operator, compared to the control. Statistics are calculated against (a) IASI observations on Metop-B, (b) ATOVS observations on Metop-C and (c) CrIS observations on NOAA-20.

with all ROMEX observations, compared to the control. Like the RMSE scorecard (Figure 1), there are forecast degradations in many quantities. This is different from the equivalent experiments at ECMWF and DWD. One alternative which was considered is whether the above formulation of the standard deviation of the forecast error affected the results. DWD calculates the standard deviation of forecast error for each verification date (Harald Anlauf, personal communication) and averages these, rather than calculating the mean-square-error and bias for the entire period. This was attempted, but made very little difference to the

results (not shown).

     When the refractivity operator is adjusted by 0.05% for $k_1$ and 3.5% for $k_2$, the results are shown in Figure 14. The results are very similar to those shown in Figure 10 in that many of the scores are greatly improved, relative to Figure 13. It is not clear why the bias corrections applied to the operator are having a clear effect on the standard deviation of the forecast error, unlike other centres.







**Figure 13.** Scorecard showing the change in standard deviation of forecast error for the experiment with all ROMEX observations, compared to the control. Figure format as in Figure 1.





**Figure 14.** Scorecard showing the change in standard deviation of forecast error for the experiment with all ROMEX observations but with a reduction by 0.05% of $k_1$ and a reduction of 3.5% of $k_2$ in the refractivity operator, compared to the control. Figure format as in Figure 1.



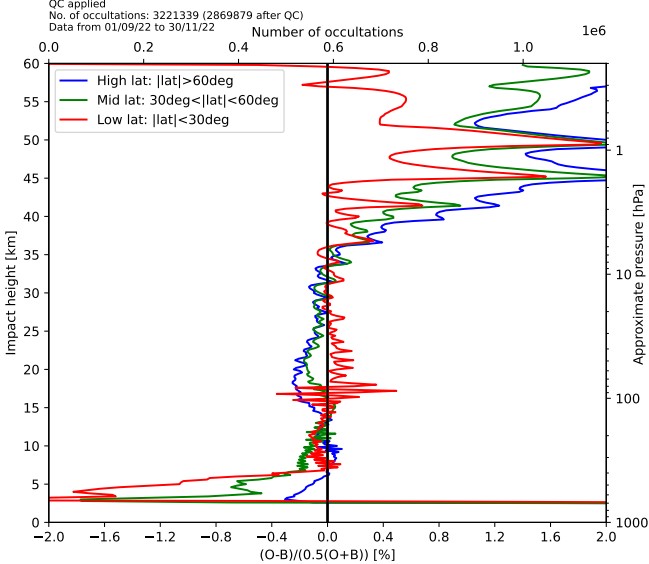

**Figure 15.** Mean difference between the observations and the model background of the observations, normalised by their average. The data are averaged over all the ROMEX observations, but binned by latitude.

## 3 Spatially varying bias

Whilst Figure 2 shows a clear picture that there is an approximately constant bias between the observations and the model background between 7 and 30 km impact height, this does not give the entire picture. Instead, these biases vary both with height and location. Figure 15 shows the variation of the mean O-B statistics with latitude. From this we note that the bias appears to vary quite considerably between the different latitude bands. In the tropics the bias in the troposphere is considerably more negative than at higher latitudes. The rapid variation in the bias in the tropics between 15 and 20 km impact height is due to the interpolation from model levels to impact heights, and therefore is an artefact of the observation operator. In the lower stratosphere there is a negative bias at mid and high latitudes, but a slight positive bias in the tropics. This is suggestive that the adjustments to the coefficients $k_1$ and $k_2$ may be too simple to properly account for the bias issues. However, we should note that these statistics make no separation between biases in the observations (which we normally assume to be small) and biases in the model background (which are often known to be large).

One further way to understand the spatial variation of the bias is to examine the spatial distribution of the O-B statistics. Figure 16 shows the mean values of (O-B)/B (note: not (O-B)/(0.5(O+B)) as before), averaged over 10 degree bins for various satellite groupings, within the vertical range 17 to 20 km impact height. This impact height range is chosen, as it is in the stratosphere at all latitudes, and experiments have shown that the lower stratosphere is where GNSS-RO observations have most impact since their errors are often smallest in this region Kursinski et al. (1997). For most of the observation groups it





will be noted that a positive O-B is seen in the tropics, and a negative O-B in the mid-latitudes. This is consistent with the results shown in Figure 15. COSMIC-2 observations are an exception to this, due to certain choices made in their processing and the impact of their low-inclination orbit connected to variations in the earth's radius of curvature with latitude (Josep Aparicio, Christian Marquardt, personal communication). It is notable that the statistics for FY-3 satellites are slightly more

negative than the others in the northern extra-tropics, and that the FY-3 and Metop satellites are slightly more positive than the others in the tropics.

### 3.1    Attempts to apply spatially varying bias correction

Given the latitudinal and longitudinal variation in the O-B statistics, it is possible that a spatially varying bias correction could be applied to the observations. This may be able to account for the variation in bias between the different observation groups,

but also for the strong variation of the bias with latitude. To test this hypothesis, the bias statistics were calculated in 30-degree latitude/longitude bins, and the mean statistics of (O-B)/B were calculated for each of the observation groups shown in Figure 16. The statistics were calculated using the following height bins: 2-5, 5-8, 8-12, 12-17, 17-20, 20-25, 25-30, 30-35, 35-40, 40-45, 45-50, 50-55 and 55-60 km impact height. Once these statistics were calculated they were used as the basis for the bias correction. When the observation were ingested, the latitude, longitude and impact height was compared with the

bin centres, and the mean (O-B)/B statistics were linearly interpolated to the observation locations. The observations were then bias corrected using these statistics. Thus, if the mean (O-B)/B statistics were equal to 0.01 at the observation location, then the observation was multiplied by 0.99 (i.e. a positive (O-B)/B indicates that the observation needs to be reduced). In this experiment we are correcting the observation for the apparent bias — irrespective of whether that bias is due to the observations or the model background.

Figure 17 shows the results of this experiment. Unlike the experiments which alter the refractivity coefficients, this experiment displays improved scores for geopotential height forecasts at 50 and 100 hPa in the northern extra-tropics when verified against ECMWF analyses. This would suggest that the bias correction at high altitudes is somewhat different to the effect of changing the coefficients. However, the RMSE for the tropospheric geopotential height forecasts are increased, suggesting that the bias correction is not performing as well as the refractivity coefficient changes. Whilst the overall score for verification

against ECMWF analyses is approximately as good as the overall score when changing both the $k_1$ and $k_2$ coefficients, the RMSE against observations is worse. This suggests that the spatially varying bias correction is not as effective as the refractivity coefficient changes, but may provide useful effects at high altitudes.

## 4    Impact of increasing the number of observations

The stated aim of the ROMEX project is to quantify the benefits of increasing the volume of RO observations by incorporating

additional data not currently available to weather centres for real-time operational systems. Therefore, it is good to assess how the quality of the NWP forecasts changes with the number of observations. If we were to perform this assessment using the unadjusted operator, then we would be measuring the degradation brought by the increase in the number of observations.





**Figure 16.** Mean difference between the observations and the model background of the observations, normalised by their average. The data are averaged over all the ROMEX observations, but binned by latitude and longitude.





**Figure 17.** Scorecard showing the change in root-mean-square forecast error for the experiment with all ROMEX observations but with a spatially varying bias correction, compared to the control. Figure format as in Figure 1.



Hence, the above corrections to the system are necessary to demonstrate the benefit which can be achieved using a well-calibrated system.

Following discussion, the following experiments have been run:

- No GNSS-RO observations

- The ROMEX control, but excluding observations from the COSMIC-2 satellites

- The ROMEX control

- A selected set of observations, with around 20,000 occultations per day

- All ROMEX observations

This range of experiments covers a wide range of observation numbers, which should therefore demonstrate a wide range of behaviour. Experiments were run with each of these datasets, using a modified version of the operator. The modifications to the operator are to reduce $k_1$ by 0.05% and $k_2$ by 3.5%, as described earlier. In addition to this, a vertical smoothing of the observations within a profile is applied where the smoothing length-scale is approximately 3/4 the model-level spacing, as

described in Bowler and Marquardt (2025). Finally, a simple bias correction is applied to high-altitude observations. For this, we define latitude regions of 0-30, 30-60 and 60-90 degrees away from the equator, and calculate approximate bias corrections for each region. The bias correction within 10 degrees of each region boundary is linearly interpolated and is applied as a multiplicative factor to the observed bending angles. Within the tropics the bias correction decreases linearly from 1 (no correction) at 36 km to 0.988 at at 48 km, and then back to 1.0 at 60 km. Between 30 and 60 degrees (north and south) the

bias correction factor decreases linearly from 1 at 34 km to 0.985 at 46 km and stays at this value above this height. Above 60 degrees the bias correction factor decreases linearly from 1 at 35 km to 0.983 at 45 km, and stays at this value above this height.

The observed bias at high altitudes is different between the ECMWF and Met Office systems. The Met Office system has a positive bias at high altitudes, whereas the ECMWF system has little bias. This suggests that the biases that are observed in the

Met Office system are unique to this system. This may be partly due to the model, but also due to the choice of deriving the virtual temperature from the pressure and humidity in the forward operator, rather than taking the temperature directly from the model. Therefore, applying this bias correction to the observations is unjustified since we are correcting the observations to look more like the model. However, applying a bias correction at high altitudes appears to be effective at improving the forecast quality, and therefore it is included in the experiments.

The dataset with 20,000 occultations per day was provided by EUMETSAT. This dataset included the following satellites and satellite constellations: COSMIC-2, Metop, TanDEM-X, TerraSAR-X, Sentinel-6A, PlanetIQ and Spire. The individual Spire satellites were chosen each day to provide a total of approximately 20,000 occultations, after quality control has been applied. Thus, there were more than 20,000 occultations each day in this dataset.





## 4.1 Trial results

Figure 18 shows the results of an experiment with all ROMEX observations, using the modified operator described above. The results are broadly similar to the experiment with altered refractivity coefficients (Figure 10), but with a few notable differences. The improvements when verifiying against ECMWF analyses are generally larger, such as the short-range forecasts in the tropics. Unfortunately, some scores are also degraded, such as the temperature at 100 hPa in the medium range. Nonetheless, this configuration was deemed to be the best performing of all the experiments, and therefore was used for the tests of the impact of increasing the number of observations.

To compare the impact of the number of observations on the forecast quality we use the overall figure from the scorecard. This is the average of the change in RMSE for all the variables and lead times shown in the scorecard. No weighting is given, so the individual components are treated equally. Figure 19 shows the change in the overall value for the RMSE scorecard for the different experiments. The results fit well to a logarithmic dependence of the overall forecast quality varying with the number of observations, as was proposed by Bowler (2020a). However, it will be noted that the forecast quality is lower for the experiment with all ROMEX observations than for the experiment with 20,000 occultations per day.

The apparent degradation when assimilating all the observations caused some confusion until it was noted that experiments testing the assimilation of FY-3E into our operational environment showed a degration in performance Lewis (2025). The 20,000 occultations per day dataset doesn't include observations from the following satellites and satellite constellations: FY-3, Tianmu, Yunyao, KOMPSAT-5 and GeoOptics. Therefore, if observations from these satellites are less beneficial, or even harming the forecast quality then we would expect the 20,000 occultations per day dataset to be the best performing. However, it should be noted that during January 2025 the Chinese Meteorological Administration introduced an update to their processing of Fengyun observations, which appears to have led to substantial improvements.

To test the hypothesis that different satellites provide different levels of benefit, a further experiment was run creating an approximately 20,000 occultations per day dataset, but with a different mix of satellites. This experiment used observations from the following satellites and satellite constallations: COSMIC-2, FY-3, Metop, KOMPSAT-5, PAZ, PlanetIQ, Sentinel-6A, TanDEM-X, TerraSAR-X, Yunyao and Spire (flight-model numbers 150, 162 and 163 only). Note that this dataset has slightly fewer observations at 20,223 observations per day, compared to 21,496 observations per day for the EUMETSAT dataset. Figure 20 shows the scorecard for this experiment, compared against the experiment with the EUMETSAT dataset. There are clear degradations in many quantities for using the alternative dataset, especially at short range in the southern extra-tropics. Therefore, it seems that the quality of the observations being used is important, and not simply the quantity.

## 4.2 Data assimilation statistics

Given that we see saturation in the RMSE scores with the addition of more observations, it is interesting to consider whether this is also seen in the data assimilation statistics. Figure 21 shows the ratio of the standard deviation of the difference between the observations and model background for the different experiments, relative to the test without GNSS-RO observations, as a function of the number of observations. For all groupings of the observations, the use of GNSS-RO observations improves





**Figure 18.** Scorecard showing the change in root-mean-square forecast error for the experiment with all ROMEX observations but with a modified operator which uses altered refractivity coefficients, vertical smoothing of the observations and a bias correction at high altitudes, compared to the control. Figure format as in Figure 1.



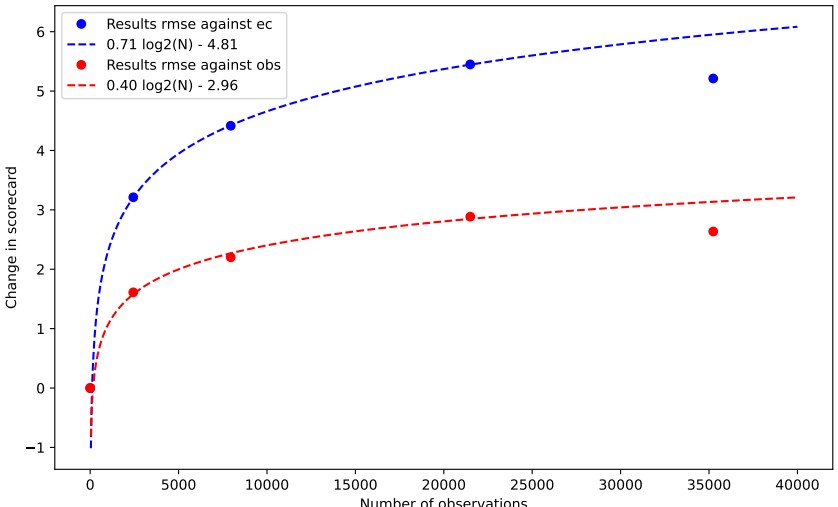

**Figure 19.** Change in the overall RMSE scorecard for the different experiments. Verification against ECMWF analyses is shown in blue, and verification against observations is shown in red. A best-fit curve based on a variation of the overall figure with the logarithm of the number of observations is shown as the dashed line. The curve has been fit to the experiments using Control - COSMIC-2, control, and 20,000 occultations per day. The number of observations shown is an approximate figure calculated before quality control.

the fit of the model background to the observations. The amount of improvement depends on the observation type being considered, which may reflect the level of noise intrinsic to the observations. The biggest reduction in standard deviation is seen for radiosonde potential temperature, which is improved by up to 4% with the addition of GNSS-RO observations. This

is the only observation type for which the all-observations experiment performs better than the 20,000 occultations per day experiment, although this difference is not statistically significant. The other observation types appear to show that the all observations experiment performs worse than the 20,000 occultations per day experiment, consistent with the overall forecast results.

### 4.3 Which variables are driving behaviour?

Since we are noting that the benefit from additional observations appears to saturate, due to the final set of observations being less beneficial in the Met Office system, it is interesting to consider which variables are driving the behaviour. Figures 22 and 23 show the change in RMSE for the different variables in the various latitude regions, as a function of the number of observations. For verification against sonde observations (Figure 22, the RMSE in the southern extra-tropics for the all-observations experiment decreases much more than for the other regions. For the northern extra-tropics there is very little

additional benefit for the addition of observations above the baseline which excludes COSMIC-2 observations. For the tropics,





**Figure 20.** Scorecard showing the change in root-mean-square forecast error for the experiment using the alternative 20,000 occultations per day dataset, compared with the EUMETSAT 20,000 occultations per day dataset. Both experiments used the modified operator which uses altered refractivity coefficients, vertical smoothing of the observations and a bias correction at high altitudes. Figure format as in Figure 1.





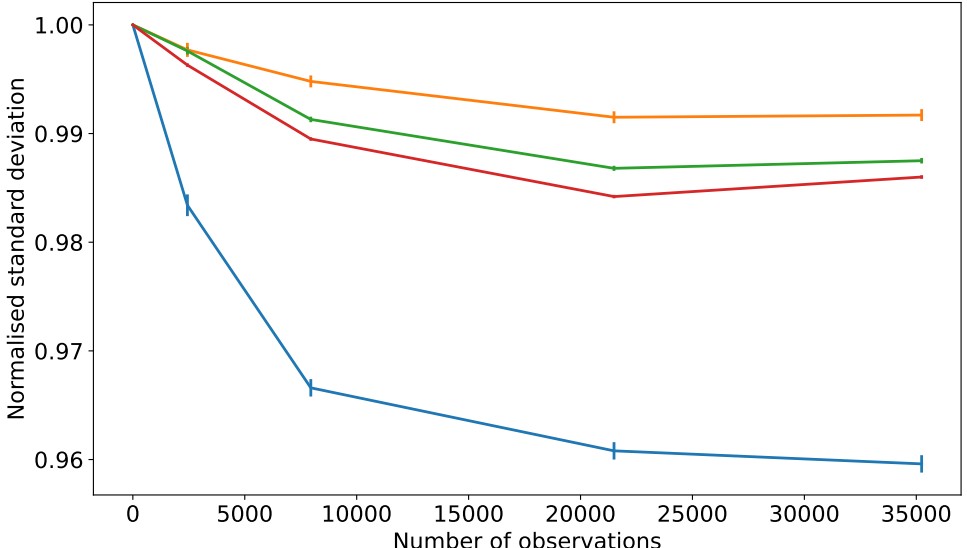

**Figure 21.** Ratio of the standard deviation between the observations and model background for the different experiments, relative to the experiment with no GNSS-RO observations, as a function of the number of observations. Shown are the statistics for upper-levels winds (orange, taken from aircraft, sonde and atmospheric motion vectors), microwave radiances (green, aggregated from ATOVS on Metop-B/C, NOAA-15/18/19, and ATMS on JPSS and NOAA-20), hyperspectral infrared radiances (brown, taken from IASI on Metop-B/C and CrIS on JPSS and NOAA-20) and potential temperature from radiosondes (blue). The number of observations shown is an approximate figure calculated before quality control. Note that the 20,000 occultations per day experiment shown here is the default one provided by EUMETSAT.

the RMSE reduction continues with the inclusion of COSMIC-2 observations, and shows limited improvements thereafter. For the southern extra-tropics the reduction in RMSE continues up to 20,000 occultations per day, but then increases with the addition of all ROMEX observations. Therefore, it is likely that the southern extra-tropics is driving the degraded performance with all ROMEX observations, but that the other regions mostly saturate at a lower number of observations.

When verifying against ECMWF analyses (Figure 23), the RMSE reduction is more uniform across the different latitude regions. Whilst the rate of improvement in the RMSE scores decreases with an increasing number of observations, the RMSE scores are improved in the tropics and northern extra-tropics with all ROMEX obsevations, relative to the 20,000 occultations per day dataset: there is no sign of saturation of benefit in the addition of observations. This is not the case for the southern extra-tropics, where the RMSE scores are degraded with the addition of all ROMEX observations. Looking at the individual

scorecards (not shown) the degradation in RMSE scores in the southern extra-tropics cannot be ascribed to a single variable or lead time, but instead is the combined effect of slight degradations in all variables and lead times.

The differences in the tropics and northern extra-tropics when verifying against ECMWF analyses and sondes are likely due to the differences in locations of the verifying data points. Sondes in the northern extra-tropics are concentrated over land, and



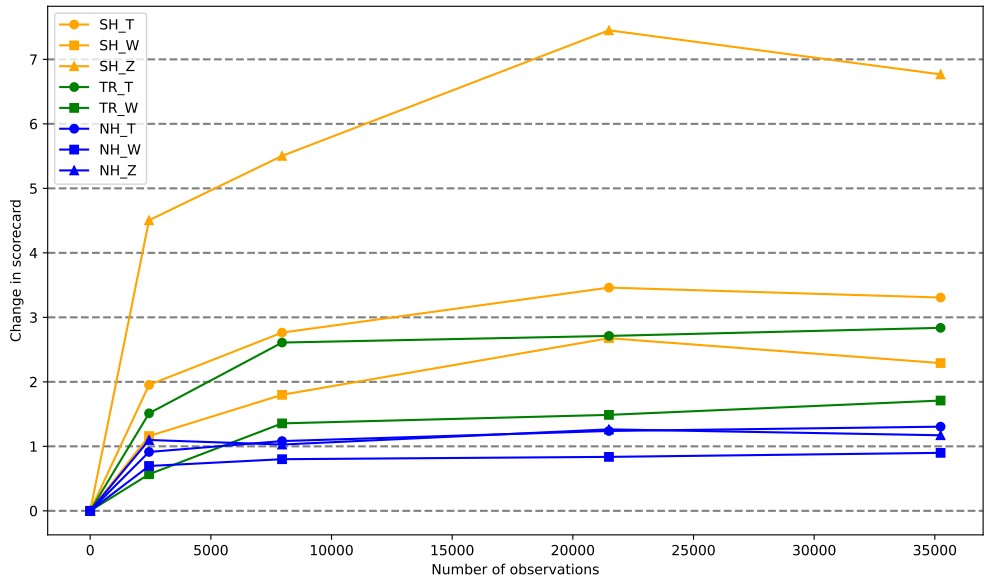

**Figure 22.** Change in the RMSE scorecard for the different variables, as a function of the number of observations for variables in different latitude regions. Verification against sonde observations. The number of observations shown is an approximate figure calculated before quality control. Note that the 20,000 occultations per day experiment shown here is the default one provided by EUMETSAT.

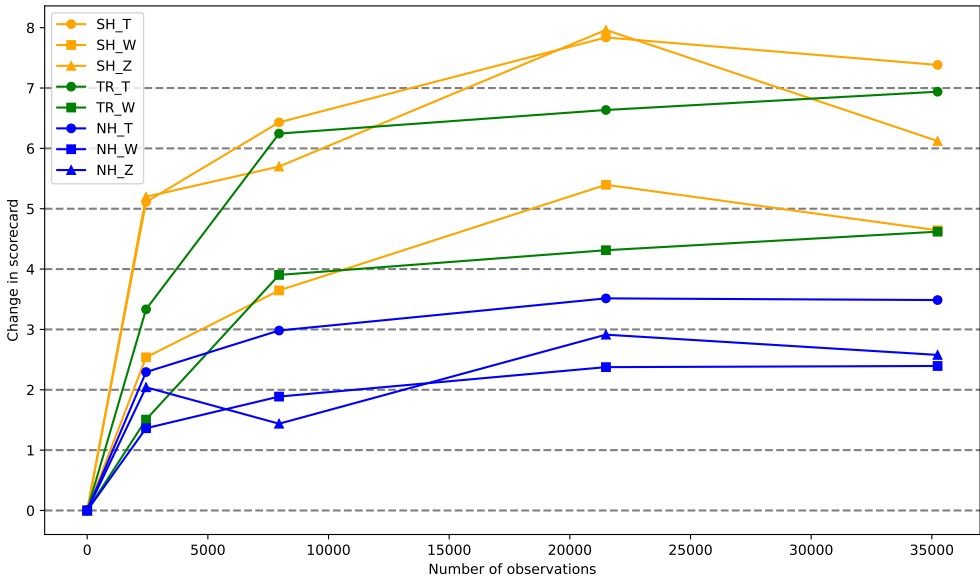

**Figure 23.** Change in the RMSE scorecard for the different variables, as a function of the number of observations for variables in different latitude regions. Verification against ECMWF analyses. The number of observations shown is an approximate figure calculated before quality control. Note that the 20,000 occultations per day experiment shown here is the default one provided by EUMETSAT.



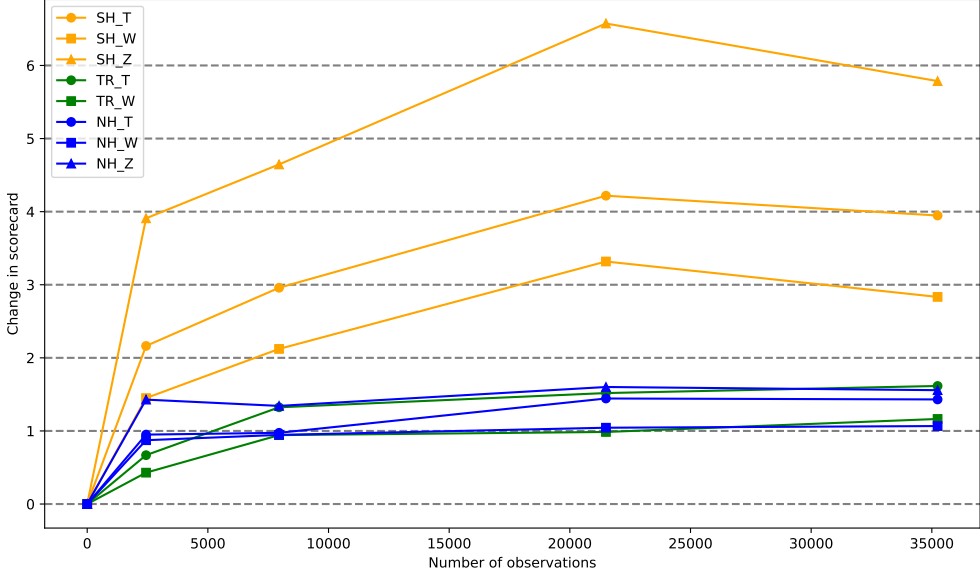

**Figure 24.** Change in the RMSE scorecard for the different variables, as a function of the number of observations for variables in different latitude regions. Only variables at 850, 500 and 250 hPa have been included in the calculation. Verification against sonde observations. The number of observations shown is an approximate figure calculated before quality control. Note that the 20,000 occultations per day experiment shown here is the default one provided by EUMETSAT.

particularly over Europe. This is a very well observed region which benefits less from additional observations. Verification against ECMWF analyses is performed on a 1.5 degree regular grid, with a weighting using the cosine of the latitude. This means that these results will be sensitive to changes in performance over the oceans as well as over land. Since the oceans are less well observed, the additional GNSS-RO observations are likely to be more beneficial.

Another factor is the behaviour of the forecast in the troposphere and at higher altitude. Figures 24 and 25 show the change in RMSE, like Figures 22 and 23, but only considering variables at 850, 500 and 250 hPa. For these lower-altitude quantities the benefit from additional GNSS-RO observations appears to continue to higher observation numbers. This is because the alterations to the refractivity coefficients have been effective at correcting the bias problems at these altitudes, whereas the bias problems at higher altitudes are not so well corrected. When verifying against observations (Figure 24), there is still early saturation of benefit in the northern extra-tropics. However, in the tropics there is no sign of saturation. When verifying against ECMWF analyses (Figure 25) the benefit in the northern extra-tropics appears to saturate at 20,000 occultations per day, but no saturation is seen in the tropics. In both cases the behaviour in the southern extra-tropics is similar to that seen in the full RMSE scorecard, with a degradation in performance with the addition of all ROMEX observations.





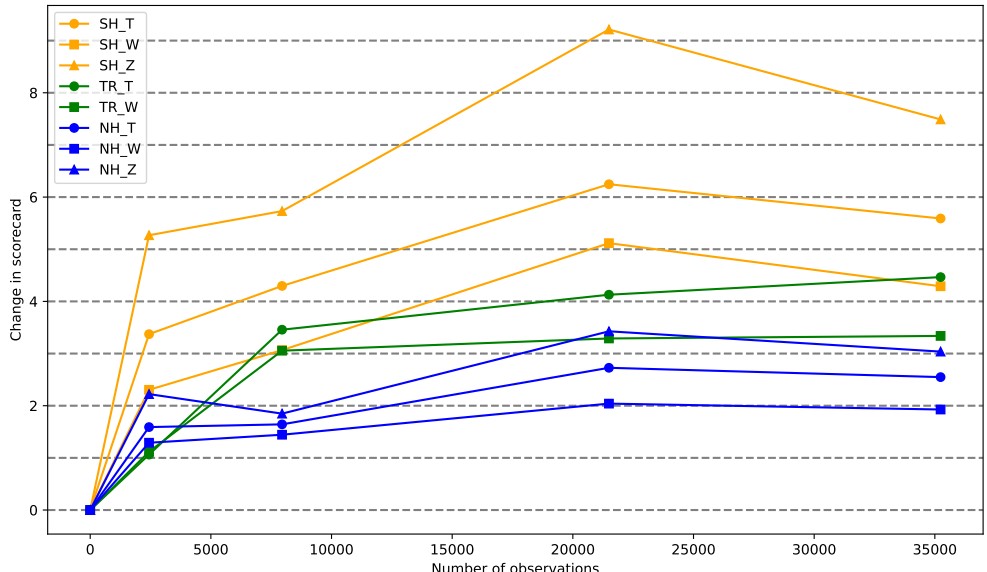

**Figure 25.** Change in the RMSE scorecard for the different variables, as a function of the number of observations for variables in different latitude regions. Only variables at 850, 500 and 250 hPa have been included in the calculation. Verification against ECMWF analyses. The number of observations shown is an approximate figure calculated before quality control. Note that the 20,000 occultations per day experiment shown here is the default one provided by EUMETSAT.

## 5 Conclusions

The ROMEX set of experiments have aimed to demonstrate the benefits to NWP from the use of additional GNSS-RO obser-
vations. Unlike previous studies which have used synthetic observations, the ROMEX project has used real observations from
a variety of satellites and satellite constellations. On the one hand, the results are more trustworthy because they are based on
real observations. On the other hand, the tests have highlighted issues with the use of the observations, and the need to adjust
the observation operator to account for forecast biases which have appeared from the use of the observations.

The initial experiments with the additional observations showed a degradation in the forecast quality, which was unexpected.
The degradation was particularly clear for tropospheric geopotential height forecasts, which was shown to be largely due to
a forecast bias. Various experiments were run exploring the effect of bias corrections to the observations, and changes to
the coefficients of the refractivity operator, which achieve a broadly similar effect. It was found that the bias in tropospheric
geopotential height forecasts is largely controlled by observations in the lower stratosphere, and small changes to the refractivity
coefficients in the observation operator are able to control this. Experiments within a 1D-Var framework demonstrated that the
change in geopotential height biases are largely due to pressure increments from observations centred around 15 and 10 km
impact height. The global average difference between the observations and background forecast in the lower stratosphere





is around 0.05%, which leads to a suggestion that this might be an appropriate adjustment to make to the dry refractivity coefficient.

Following these experiments it was found that changes to both the $k_1$ and $k_2$ coefficients in the refractivity operator performed best, although neither change has a clear theoretical justification. It was decided to reduce $k_1$ by 0.05% and $k_2$ by
3.5%. This was shown to be effective in reducing the forecast bias, and improving the forecast quality, except when verifying geopotential height forecasts at 50 hPa against ECMWF analyses. Therefore, these changes were used in the final experiments. Other centres have shown similar levels of bias when assimilating the full set of ROMEX observations. However, whilst the Met Office forecasts were degraded by this bias, both when verifying with the RMSE and the standard deviation of forecast error, other centres found that the standard deviation of forecast error was improved, with or without changes to the refractivity
coefficients. It is not clear why the Met Office stands out as being particularly negatively affected by the bias changes.

The apparent bias between the observations and the model background varies spatially, in all three dimensions. The apparent bias is largely similar between different groups of observations, which may suggest that the biases are dominated by model errors rather than deriving from the observations. However, there are certain differences, such as a positive bias in the COSMIC-2 observations which is becoming better understood. Applying a spatially-varying bias correction to the observations, based
on the measured apparent bias was attempted. This proved to be effective in many respects, particularly when verifying against ECMWF analyses. However, the overall performance was not as good as the refractivity coefficient changes, and therefore this approach was not pursued further.

For the final experiments a configuration was chosen which used adjusted values of the refractivity coefficients, applied a latitude-dependent bias correction for observations at high altitude (to correct for large model biases in this region), and used a
vertical smoothing of the observations. With these settings, experiments were run with a varying number of observations. It was found that the overall RMSE scores improved with increased numbers of observations, up to a maximum of 20,000 occultations per day. Beyond this point, the RMSE scores were slightly degraded, particularly in the southern extra-tropics. It was found that the 20,000 occultations per day dataset preferentially removed certain satellites and satellite constellations. Some of these satellites have been shown independently to be less beneficial to the NWP forecasts from the Met Office, and therefore the
particularly good performance of the experiment with the 20,000 occultations per day dataset is likely due to its inclusion of only satellites which are highly beneficial in the Met Office system. When splitting the results by latitude region, it was found that the southern extra-tropics was driving the degradation in performance with the addition of all ROMEX observations.

It should be noted that the benefit of the additional GNSS-RO observations was only demonstrated in the Met Office system once the adjustments to the refractivity operator were made, correcting the forecast bias that was apparent. Experiments with
a more sophisticated derivation of the refractivity operator were not conducted. However, similar experiments made at other centres demonstrated similar changes in the bias, and therefore it is not guaranteed that a different operator would yield different results. The appearance of these issues only occured because the experiments used real, rather than synthetic, observations, which demonstrates the value of using real observations in the experiments.





*Author contributions.* The experiments were designed by NEB. The data preparation and experiment runs were jointly conducted by NEB
and OL. NEB ran the verification, interpreted the results and prepared the manuscript.

*Competing interests.* None of the authors have any competing interests.

*Code and data availability.* The ROMEX RO profiles are provided through EUMETSAT ROM SAF. The code is not publicly available.



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
