# Peer review of "Experiments with large number of GNSS-RO observations through the ROMEX collaboration in the Met Office NWP system"

_EGUsphere, 2025_

## Referee Comment (RC2)

**Review of "Experiments with large number of GNSS-RO observations through the ROMEX collaboration in the Met Office NWP system"**

By Neill Edward Bowler and Owen Lewis

This manuscript presents a study on the impact of a large number of GNSS-RO data provided by the ROMEX project on the Met Office's NWP system. The introduction is concise and very well written. It provides a clear and easy-to-follow overview of the GNSS-RO observations and the ROMEX project. The authors have done substantial work on this topic and cover many aspects in the assimilation of GNSS-RO observations from ROMEX. The results are solid and inspiring.

I feel that the presentation could be improved so that it can be more easily followed by readers from a broader community. Particularly, the authors use "to test the hypotheses" quite a few times to introduce their new experiment. This makes the presentation flow hard to follow sometimes. I suggest that the authors include a table listing all the experiments and their major configuration parameters before discussing the results, or alternatively, place the table in an appendix. Otherwise, I have only a few minor comments below.

**Specific comments:**

L1, GNSS-RO should be spelled out.

L 29, 39, 44,47, 56, 156, and many others....

I do not know the specific formatting requirements of AMT, but having references outside the parentheses seems weird. For example, in L29: "NWP forecasts have been gradually improving in quality over time **Bauer et al. (2015)."**

L 40-42, I'm uncertain about the relationship between WMO and CGMS, but this phrasing looks questionable to me. Please verify.

Fig. 1,

- a) why are TR Z forecasts missing in the scorecards? Is there any reason for this?
- b) The right panel of the scorecard figures shows verification against observations. Could the authors provide more details about this? Specifically, what types of observations are being referred to?

L133: Looking again at 2 -> Fig. 2

Fig. 6, The figure was wrong in the original submission. While I can access the new figure attached in the Editor's comment, the authors need to include the new one in the revision. It is true that "The experiment which adjusts the observations by 0.05% approximately halves this reduction, and the experiment adjusting by 0.1% eliminates it entirely. This

seems to be the main reason that the adjusted experiments perform better than the initial experiment". However, the forecast bias of the adjusted experiments increases with lead time. Please explain.

Fig. 7, Could the authors adjust the y-axis range for the humidity panel to improve visualization, given that there is nothing significant above 10–12 km?

Fig. 9, the paper presents many scorecard figures to show the overall impact of a test. As stated from L194 "There is evidently strong similarity between the results of this experiment and the ones shown in Figure 5, although not exactly equivalent." I do not think Fig. 9 is necessary to be shown. A summary sentence could be enough. The authors can consider remove Fig.18 for similar reason too.

L 236, spell out DWD please.

Fig. 16a, please provide more explanation on "COSMIC-2 observations are an exception to this,..."

L348, adding a reference could be helpful.

Eqs. 3-5, I do not think these equations are necessary given these are standard statistics, but I am fine with them included.

---

## Editor Comment (EC1)

95% confidence intervals calculated from SE assuming independent observations

---

## Author Comment (AC2)

**Reviewer Comment 4**

Thank you for your review. The original text is maintained in black, whereas our responses are marked in red. Any changes to the revised manuscript are also indicated in red text.

This article "Experiments with large number of GNSS-RO observations through the ROMEX collaboration in the Met Office NWP system" by Bowler and Lewis discusses the impact by increasing GNSS-RO observations during ROMEX on numerical weather prediction. First, it was stated that forecast scores were degraded caused by stratospheric effects which required adjustments e.g. to the refractivity operator and data processing. After these corrections, assimilating more GNSS-RO data—particularly around 20,000 observations per day—greatly improved forecast accuracy, especially in the southern-hemisphere extra-tropics.

This manuscript highlights nicely the potential but also the challenges coming from assimilating a high number of GNSS-RO data, which wasn't done before. The authors looked into many different aspect, e.g. bias correction and changes to the forward operator to address the challenges. I recommend publishing these results after major revision.

**Main points:**

In general, I think the manuscript needs to be slightly trimmed and restructured perhaps. e.g. the explanation of RMSE/ bias is done after using this measure in various score cards and with using only observations as a reference. Also, many sensitivity studies have been done to tackle the bias in geopotential. Rather showing all the different scorecards and only have a little discussion about them, I would focus on the most important ones but keep mentioning the results obtained from the various experiments with e.g. bias correction. I truly believe this makes the manuscript more readable.

> Following the suggestions of other reviewers, we have removed Figure 9, and have added a table summarising the list of experiments which are presented. This table includes references to the section in which the experiment is discussed, which will hopefully allow the user to navigate the paper more easily.

Also, I am a bit reluctant to accept the main conclusion that there is a saturation in impact for the 20.000 daily occultations. As the authors discussed, an alternative flavour of this sample showed lower forecast impact, which could cause different fits as previously. This needs to be shown and discussed referring to a possible impact of data quality. However, here I would be careful not to only make the quality of FY3E responsible for that behaviour without making an additional analysis. Nevertheless, differences in timing of the observation, geographical coverage and quality of the observations are playing a key role in their impact.

> We feel that this study shows that the quality of the observations is important, and one cannot treat all observations as equal. We did not intend to suggest that one satellite (or constellation) is responsible for the apparent saturation of benefit and have reworded that discussion appropriately.

However, our experiments have shown that we see limited benefit from the additional observations above the control in certain (well-observed) regions. We have avoided using the word "saturation" in either the abstract or conclusions, as this word can carry certain unwanted implications.

**Other points**

Missing brackets for citing other literature throughout the paper

This error derives from the history of the manuscript (it was originally written in a format which only allowed the use of the \cite command). This has now been fixed.

P1, abstract: I find it confusing to read first about negative impacts and then later substantial improvement in forecast quality. Please make it clearer what changed to get better forecast with more RO data.

We have added an additional sentence near the start of the abstract to highlight be improvements before going on to address the challenges. The paragraph in question now begins:

After making various changes to the observation operator used to assimilate the ROMEX observations it was found that the additional observations made substantial improvements to the forecast quality. Without these changes the additional data was seen to degrade the forecast quality, highlighting the importance of understanding the biases within the NWP system. The negative impacts were largely due...

P1, l21/22: This sounds like a hypothesis which has not been analysed in this study - hence, I would avoid stating that.

This is a reference to the experiments which constructed an alternative version of the 20,000 occultations per day dataset, and demonstrated smaller benefits with this. We have reworded this to be more explicitly linked to our results:

Overall the largest forecast improvements were seen when assimilating 20,000 occultations per day. An alternative dataset was also created with 20,000 occultations per day, but with a different choice of satellites. This alternative dataset gave smaller benefits than the official one, indicating that the quality of the data from each satellite is also important.

P3, l78: typo in observation

Thank you, corrected.

P4, l91 Would add "horizontal" before "resolution"

Done.

P 4, Section 2.1: Please state which metric is being looked at. RMSE?

We've updated this first sentence to read:

A summary of the changes in the RMSE scores for various variables and forecast lead times for the initial experiments are shown in Figure 1.

P4, l.105: How big is "large". Please quantify.

We have brought forward the discussion of the forecast biases (promoting Figure 6 to be Figure 2). The discussion of biases is now split between Section 2.1 (where the initial experiments are discussed) and Section 2.2 (which applies a bias correction to the observations).

P5, l120-123: What do you mean with adjusting the observations? Modifying the bending angles which are assimilated or the corresponding impact? parameter?

Yes, we have modified the observed bending angles. We have modified the wording to make this clear:

If we modify the observed bending angles in this region to be larger, then the bias in O-B in this region will be reduced. An experiment was run where the observed bending angles were adjusted by a factor, linearly increasing from 1 (no adjustment) at 7 km impact height, to 1.025 at 0 km impact height (noting that this is below the earth's surface).

P8, l137 – same as in previous comment. What do you increase here?

As before, we have changed "observations" to "observed bending angles".

P12, Figure 12: What is pert_all? A perturbation of bending angles in all height levels or an average of pert_5km and so on. If the latter, how can pert_all be positive but the individual impact heights are mostly negative?

The perturbation is to the bending angle at all heights. The pressure increment in Figure 7 is indeed positive for pert_all, whereas the pressure increment for the pert_{height} experiments is mostly negative. This is easiest understood from Figure 8, which looks at the change in the increment as a result of the perturbations. This shows that the effect of the perturbation is to increase the pressure in all the tests, with the biggest increase for pert_all. Since the unperturbed increment to the pressure is negative, these increases are being added to a negative baseline.

P12, L178: Which change? The change due to adding ROMEX compared to the control?

We have updated this sentence to be clearer:

The above results indicate that the reduction in the short-range geopotential height forecasts when assimilating the additional GNSS-RO observations is likely due to a systematic reduction in the atmospheric pressure.

P 13, l190: "may be preferred" sounds a bit out of place here. Maybe typo?

This is a bit of loose language. We have improved the sentence to read:

The constants used in this equation are derived from rather old experiments, and more recent formulations of the refractivity may may lead to more accurate simulations of the atmosphere (Healy, 2011; Aparicio and Laroche, 2011).

P15, l 200-204: Here or even better earlier it would be good to discuss if this change in the mean/bias is solely or partly responsible for degradations seen in RMSE

(scorecards). What about std deviation? Does this also contribute to an increase in RMSE?

> We have included a forward reference to the changes in the bias, so that the reader understands that these topics will be discussed.

P15, Eq (3): This would be good to define earlier in the manuscript.

> It is hard to see an alternative location where this would fit, so we have left the equation in its current location.

P17, Eq(4): I would rename $o_{i,t}$ as truth or reference, which can be observations but in other instances analysis.

> We have adjusted this term to "verifying reference value".

P18: This part jumps a bit from verifying against observations and against analysis - hence I'd define standard deviation and rmse with using reference or truth rather than observations.

> Yes, the use of observation was an oversight.

P25, l 314: remove one of the "at"s

> Deleted.

P25, paragraph l 318-324: It would be good to see the impact in forecast scores for that experiment.

> Experience tells us that changes to observations at high altitudes tend to have relatively small impacts on the NWP forecast skill. This is partly because the main NWP forecast quantities are in the troposphere and partly because the DA system gives low weight to high-altitude RO observations. One experiment that we have run which is not included in the paper is to bias correct observations below 34 km impact height by -0.05%, and observations above 47km by +1.2%, with a linear variation between. Compared to the experiment only perturbing observations by -0.05%, this experiment had relatively limited and somewhat mixed impacts (see figures below).

P26; paragraph l.342-349: This suggests that FY3E caused the degradation. Can this be said or are the results only suggestions it could be due to different "quality"? Looking at Fig 20 I'd said the latter. Could Fig 19 be edited including the alternative 20k sample?

Figure 19 has been updated to include the alternative 20k experiment.

It was not our intention to suggest that FY-3E was the principal culprit in the behaviour that we observed.  Therefore, we have reworded the paragraph to put this mention closer to the end:

The apparent degradation when assimilating all the observations caused some confusion. However, we noted that the 20,000 occultations per day dataset is not a random sampling of the full dataset, but excludes certain satellite constellations. It excludes observations from the following satellites and satellite constellations: FY-3, Tianmu, Yunyao, KOMPSAT-5 and GeoOptics. Therefore, if observations from these satellites are less beneficial, or even harming the forecast quality then we would expect the 20,000 occultations per day dataset to be the best performing. Separate experiments testing the assimilation of FY-3E into our operational environment showed a degradation in performance (Lewis, 2025). Therefore, we speculate that different observations are of different quality and that the 20,000 occultations per day dataset kept those which are most beneficial to the Met Office's NWP system. However, it should be noted that during January 2025 the Chinese Meteorological Administration introduced an update to their processing of Fengyun observations, which appears to have led to substantial improvements (Yan Liu, personal communication).

P28, l370-371: Is this still valid to say after showing that the alternative sample in the 20k experiment did show a different behaviour? Hence, using the alternative 20k experiment might give you a different answer.

We have rephrased this sentence to:

Since we are noting that the Met Office system gains little additional benefit from the full set of observations due to the final set of observations being less beneficial in the Met Office system, it is interesting to consider which variables are driving the behaviour.

P33, l408: Please, mention in which forecast score metric this degradation can be seen and why it was unexpected.

We have rephrased this sentence to be more explicit:

The initial experiments with the additional observations showed an increase in the RMSE of forecast error. This was unexpected as one would expect the additional information provided by the observations to improve forecast quality.

P34, l424: I would rephrase this part of the sentence "standard deviation of forecast error was improved". I'd rather say that compared to the reference the standard deviation decreased, which can be interpreted as an improvement or something similar.

This has been rephrased to "the standard deviation of forecast error was reduced when assimilating the additional observations".

---

## Author Comment (AC3)

**Reviewer Comment 2**

**Review of "Experiments with large number of GNSS-RO observations through the ROMEX collaboration in the Met Office NWP system"**
By Neill Edward Bowler and Owen Lewis

Thank you for your review. The original text is maintained in black, whereas our responses are marked in red. Any changes to the revised manuscript are also indicated in red text.

This manuscript presents a study on the impact of a large number of GNSS-RO data provided by the ROMEX project on the Met Office's NWP system. The introduction is concise and very well written. It provides a clear and easy-to-follow overview of the GNSS-RO observations and the ROMEX project. The authors have done substantial work on this topic and cover many aspects in the assimilation of GNSS-RO observations from ROMEX. The results are solid and inspiring.

I feel that the presentation could be improved so that it can be more easily followed by readers from a broader community. Particularly, the authors use "to test the hypotheses" quite a few times to introduce their new experiment. This makes the presentation flow hard to follow sometimes. I suggest that the authors include a table listing all the experiments and their major configuration parameters before discussing the results, or alternatively, place the table in an appendix. Otherwise, I have only a few minor comments below.

> Adding a table of experiments is a very good suggestion. We have added such a table at the start of the section discussing the results. In this table we have included the headline scores (i.e. the average change in the RMSE for that experiment).

**Specific comments:**

L1, GNSS-RO should be spelled out.

> Done.

L 29, 39, 44,47, 56, 156, and many others….

I do not know the specific formatting requirements of AMT, but having references outside the parentheses seems weird. For example, in L29: "NWP forecasts have been gradually improving in quality over time **Bauer et al. (2015).**"

> This error derives from the history of the manuscript (it was originally written in a format which only allowed the use of the \cite command). This has now been fixed.

L 40-42, I'm uncertain about the relationship between WMO and CGMS, but this phrasing looks questionable to me. Please verify.

> The CGMS is an independent organization, and WMO members may make use of its recommendations. This paragraph has been updated to:

> The Coordination Group for Meteorological Satellites (CGMS) provides recommendations on the number of observations that should be made each day

by the various observation platforms. Whilst the CGMS is not able to mandate the number of observations to be made, it does provide guidance to national meteorological centres and world meteorological organisation (WMO) members on the number of observations that should be made.

Fig. 1,

    a) why are TR Z forecasts missing in the scorecards? Is there any reason for this?

    Geopotential height is not a useful quantity in the tropics. Therefore, it is omitted from all Met Office scorecards.

    b) The right panel of the scorecard figures shows verification against observations. Could the authors provide more details about this? Specifically, what types of observations are being referred to?

    The scorecards are calculated against radiosondes. The word radiosonde has been added to the caption of Figure 1.

L133: Looking again at 2 -> Fig. 2

    Fixed

Fig. 6, The figure was wrong in the original submission. While I can access the new figure attached in the Editor's comment, the authors need to include the new one in the revision. It is true that "The experiment which adjusts the observations by 0.05% approximately halves this reduction, and the experiment adjusting by 0.1% eliminates it entirely. This seems to be the main reason that the adjusted experiments perform better than the initial experiment". However, the forecast bias of the adjusted experiments increases with lead time. Please explain.

The error with Figure 6 has been corrected in the latest revision. We were perhaps imprecise in our explanation of this figure – it is the short-range forecast bias in 500 hPa height which is approximately halved by the 0.05% experiment. We have adjusted the text to hopefully be clearer:

The experiment which adjusts the observations by 0.05% approximately halves the negative bias in the short-range forecasts of this quantity. The experiment adjusting by 0.1% entirely eliminates the negative bias replacing with a slight positive bias, similar to the control NWP system. With increasing lead time, the forecast tends towards a positive 500 hPa geopotential height bias. The change of bias in the short-range forecast} seems to be the main reason that the adjusted experiments perform better than the initial experiment – they are able to remove the large negative bias in the geopotential height forecasts.

Fig. 7, Could the authors adjust the y-axis range for the humidity panel to improve visualization, given that there is nothing significant above 10–12 km?

Thank you for the suggestion – we have revised the humidity panel to only plot up to 15km for Figs 7 and 8.

Fig. 9, the paper presents many scorecard figures to show the overall impact of a test. As stated from L194 "There is evidently strong similarity between the results of this experiment and the ones shown in Figure 5, although not exactly equivalent." I

do not think Fig. 9 is necessary to be shown. A summary sentence could be enough. The authors can consider remove Fig.18 for similar reason too.

We have removed Figure 9, and updated the text to give the following summary:

To run an experiment equivalent to the ones above, $k_1$ was reduced by 0.1%. The results are very similar to those shown in Figure 5, and are therefore not included here.  Changing the $k_1$ value gives slightly better forecasts of extratropical temperature and wind, but slightly worse forecasts of extratropical geopotential height.  This highlights that a bias correction to the observations can have a very similar effect to an adjustment to the observation operator.

We feel that the differences between Figure 18 and Figure 10 are sufficient to justify its inclusion in the paper.

L 236, spell out DWD please.

We have added definitions for DWD and ECMWF.

Fig. 16a, please provide more explanation on "COSMIC-2 observations are an exception to this,…"

Unfortunately, a limitation has been discovered in the ROM SAF monitoring software (which is written by the Met Office and used to produce graphs on the ROM SAF NRT monitoring pages). This software, as originally written, does not account for the drifting tangent point within a GNSS-RO profile. This only has a small effect on the calculated biases for most satellites, but is important for the COSMIC-2 satellites, due to their low-inclination orbit. Therefore, Figure 16 has been updated to use the corrected software which accounts for the drift in the tangent point of the observation.  However, we have added an additional figure to demonstrate the difference that this limitation has in the case of COSMIC-2, as the experiments were run based on biases which were calculated with the original system.

L348, adding a reference could be helpful.

Unfortunately, we don't have a reference for this – we are told that a paper is in preparation on the upgrade.  We have added (Yan Liu, personal communication) to note this.

Eqs. 3-5, I do not think these equations are necessary given these are standard statistics, but I am fine with them included.

We feel that they're worth including. One justification of this is to note that the standard deviation of forecast error is derived from the RMSE, rather than the square of differences from a mean. These are equivalent in a machine with infinite precision, but can lead to issues when accumulating very large amounts of data.

---

## Author Comment (AC4)

**Reviewer Comment 1**

Thank you for your review. The original text is maintained in black, whereas our responses are marked in red. Any changes to the revised manuscript are also indicated in red text.

The manuscript provides impact assessment of massive GNSS-RO observation on UKMO's NWP system. The first half of the manuscript discusses the cause and solution of the forecast score degradation due to the introduction of large volumes of data, while the latter half presents how forecast score changes with the increasing number of observations. Observing System Experiments (OSEs) show that simply increasing the number of observations degrades forecast scores especially at troposphere. Sensitivity experiments employing 1D-Var show that observations at lower stratosphere induce change in the troposphere. OSEs with modulated refractivity observation operator coefficients, which aims to reduce the observation- background biases, show improved performance as the number of assimilated observations increase.

The manuscript provides comprehensive analysis on the impact of massive GNSS-RO observation on NWP system which also gives insight into essential role of GNSS-RO observation. The manuscript is definitely very dense and rich in information based on numerous OSE results. I think the content of the manuscript is very variable to the community and suggest publication after some revisions that might help to clarify the following points:

1. It seems that Figure 6 and its description/discussions are not consistent. Please provide more explanation on the figure and how you interpretate it.

   Unfortunately, the wrong graphic was included in Figure 6 (it was a duplicate of Figure 7a). This error has been corrected in the latest manuscript.

2. While Figure 22 to 25 shows RMSE rises as observation number increases, section 4.3 states that RMSE decreases as observation number increases. It is confusing and I suggest that taking consistency between the explanations and the figure. Also please add explanations on "change in the RMSE scorecard".

   As with the "overall" figure that is printed at the top of each scorecard, the values plotted in Figures 22 to 25 are positively-oriented (a reduction in the RMSE gives a positive overall figure). We accept that this is not clearly explained in the manuscript, so we have changed the presentation to be negatively-oriented, so that we are plotting the percentage reduction in RMSE. The figures have been changed and we feel this makes a lot more sense within the text.

3. In section 2.4 (line 178), it is stated that "change in tropospheric geopotential heights are likely due to a systematic reduction in the atmospheric pressure", whereas section 2.2 (line 130) states that "An adjustment to the atmospheric state below the observation will alter the modelled height of the observation". Since these sentences both explain same thing, ensuring consistency among these sentences will make it easy for the reader to follow the argument

The final sentences have been reworded to refer back to the hydrostatic tail which was introduced in the earlier argument:

The above results indicate that the reduction in the short-range geopotential height forecasts when assimilating the additional GNSS-RO observations is likely due to a systematic reduction in the atmospheric pressure. This is caused by observations in the upper troposphere and above, due to the hyrdostatic tail present in the forward operator (Eyre, 1994; Bauer et al., 2014).

Other suggestions:

1. line 40: CGMS is not a sub-body of WMO.

   Apologies, this has now been corrected to:
   The Coordination Group for Meteorological Satellites (CGMS) provides recommendations on the number of observations that should be made each day by the various observation platforms. Whilst the CGMS is not able to mandate the number of observations to be made, it does provide guidance to national meteorological centres and world meteorological organisation (WMO) members on the number of observations that should be made.

2. line 105. Figure 1 does not provide information about bias change. It is beneficial to present figure on bias change for the reader to follow the argument.

   We have brought forward the discussion of the forecast biases (promoting Figure 6 to be Figure 2). The discussion of biases is now split between Section 2.1 (where the initial experiments are discussed) and Section 2.2 (which applies a bias correction to the observations).

3. line133: Looking again at 2 => Looking again at **Figure**.2

   Thank you, fixed.

4. line 134: Add citation for "golden region".

   We have added a reference to Anthes et al., 2025 which explains the term.

5. Section 2.4 : It would beneficial for the reader to present figure how O-B statistics changes by modulating forward operator coefficient.

   We have produced the figure below which gives the effect of changing the coefficients in the observation operator. Unfortunately, this is only from one month of the ROMEX statistics, due to limitations on the files that we have available.  Given the already long nature of the paper, we suggest not including this.  Instead we have added a section of text describing these results.

[Figure]

BA Global O-B statistics for all satellites provided by multiple centres

**References**

Anthes, R., Sjoberg, J., Starr, J., and Zeng, Z.: Evaluation of biases and uncertainties in ROMEX radio occultation observations, Atmospheric Measurement Techniques, 18, 6997–7019, doi: 10.5194/amt-18-6997-2025, 2025.

---

## Author Comment (AC5)

**Reviewer Comment 3**

Thank you for your review. The original text is maintained in black, whereas our responses are marked in red. Any changes to the revised manuscript are also indicated in red text.

The manuscript presents a nice study on the impact of large numbers of real GNSS-RO data on the MetOffice NWP system in the context of the ROMEX project and shows some interesting lessons that can be learned with such a large dataset. It is well readable and mostly very clear, nevertheless it might need explanations in some places to address readers outside of the radio-occultation community or who are not familiar with NWP systems or data assimilation for NWP.

Futhermore the manuscript could be slightly improved by addressing the minor issues and needed clarifications as discussed below.

- Page 2, line 40: CGMS is not a subgroup of WMO, but a "multi-lateral coordination and cooperation across all meteorological satellite operators in close coordination with the user community such as WMO, IOC-UNESCO, and other user entities" (text taken from https://cgms-info.org/about-cgms/)

> Thank you for pointing this out. This paragraph has been updated to:
>
> The Coordination Group for Meteorological Satellites (CGMS) provides recommendations on the number of observations that should be made each day by the various observation platforms. Whilst the CGMS is not able to mandate the number of observations to be made, it does provide guidance to national meteorological centres and world meteorological organisation (WMO) members on the number of observations that should be made.

- Page 4, line 90ff: the description of the NWP system could be slightly more specific. Analyses every 6 hours? Which forward model used for GNSS-RO? There are several implementations, and technical details like refractivity expression etc. are of interest. (It seems clear to me that bending angle is used, but some centres use refractivity.) A reference, ideally also outlining the modeling of observation error would be great.

(Section 2.4 does discuss refractivity in passing.)

> This paragraph has been greatly expanded to include more details of the modelling system. It now reads as follows:
>
> All the experiments documented in this report were run using the low-resolution version of the Met Office's global NWP trial workflow. The resolution of the NWP forecast is described as N320, meaning that it has 640 by 480 grid points, corresponding to a resolution of around 40 km in the mid-latitudes. The model also uses 70 levels in the vertical, stretching from 20 metres above the surface to 80 km altitude. The forecasts are for the atmosphere only, and take a prescribed sea-surface temperature from the OSTIA data assimilation system (Fiedler et al., 2019). The data assimilation system is a hybrid 4D-Var system (Rawlins et al., 2007; Clayton et al., 2013), meaning that a portion of the background-error covariances are derived from the operational ensemble. The system is run in

"uncoupled" mode, meaning that an ensemble forecast is not run as part of the experiments, but the ensemble information is taken from the archive of the operational system. The data assimilation system runs on a 6h cycle, ingesting observations from a wide variety of sources, including GNSS-RO, microwave and infrared radiances, atmospheric motion vectors, conventional observations and many more. The Met Office's forward operator for GNSS-RO bending angles (Burrows et al., 2014; Burrows, 2014) is a one-dimensional operator. As discussed later, the formulation of Smith and Weintraub (1953) is used in the calculation of the atmospheric refractivity. The observation uncertainties are assigned according to Bowler (2020b), which also gives a brief description of the quality control procedures used for GNSS-RO. For satellites not available at that time an appropriate observation uncertainty estimate is used (Metop-C uncertainties are copied from Metop-A, FY-3E from FY-3C and so on). All other GNSS-RO uncertainties are taken from the COSMIC-1 estimates, although commercial GNSS-RO observations are assumed to have a 6μ rad minimum uncertainty, rather than 3μ rad as is used for most satellites.

Did the authors make initial assumptions about data quality from different missions? Some NWP centres choose not to assimilate all data equally, and the authors themselves noted (line 342) that using data from a particular mission showed a degradation of the scores.

Each satellite uses different observation uncertainties, but otherwise they are treated equally.

Line 91: Not really important, but does the lowest model layer have a full-level height of 20m, or is it the actual depth of that layer?

The Met Office's model uses a staggered vertical grid. The heights of the "theta" levels (which hold the mass variables) start at 20 m above ground and smoothly proceed to 80 km above the geoid. The wind and pressure levels start at 10 m above ground and stop at approximately 76 km above the geoid (although there is also a fictious pressure level at approximately 84 km).

- Line 102ff: when presenting forecast scores, always specify against which "truth" the verification is performed. Neither the text nor the caption of figure 1 explain the observation type being used (radiosondes)? The caption also refers to surface observations (SYNOP), but I cannot find related entries in the figure.

The observations used are radiosondes and this has been added to the figure caption. The reference to 2m temperature and 10 wind was an error, and these have been deleted.

- Line 116 and figure 2:

— does the figure show the mean error for all observations processed, or has a quality control been applied?

This is after quality control, so text has been added to note this.

— there are several spikes (or wiggles) of varying amplitude visible in the plot.  What is the origin of these spikes?  Interpolation of observations or of model equivalent?  E.g. the region between 15 km and 20 km is so noisy that the reader can hardly guess the average bias. The spikyness also varies a lot with latitude band (fig.15).  Can this have an effect on the results shown in fig.16?

These spikes are caused by the interpolation of atmospheric variables from model levels to the observation impact heights. They were reduced by the work of Burrows et al., (2014) but were not eliminated. They may have an effect on the distribution of biases which are seen in Figure 16, but not on the cause of COSMIC-2 being an outlier. That was caused by the monitoring system (from which the plots were produced) failing to account for the drift of the tangent point of each observation within a profile. That issue is now described in the appropriate section.

— does this spiky behavior also occur in the forward operator used in the variational assimilation?  If so, does it affect the modeling of error (background / observation)?

Yes, this is an interpolation issue in the forward operator which is being used and therefore affects the variational assimilation.  The observation uncertainties diagnosed from the Desroziers method (Bowler, 2020) contained these oscillations and so explicit smoothing was applied to the uncertainties.  We have added the following text to the paper:

Between 15 and 20 km there is a small-scale oscillations in the statistics. This is due to the interpolation of the atmospheric quantifies from model levels to the impact heights of the observations. These oscillations were reduced by the use of pseudo-levels (Burrows et al., 2014), but were not fully eliminated.

- Page 8, line 134: the "golden region" may be colloquially known to experts in the field, but either a reference or a less colloquial description would help the non-experts.

We have added a reference to Anthes et al., 2025 which explains the term.

- Line 148: "This shows a large reduction in the forecasts ..."
  The reduction is shown here for *bias*. It appears that geopotential bias is globally averaged (without stating so).  Is this effect seen similarly in the extratropical hemispheres? Regarding the quantification of the reduction: this is against ECMWF analyses!  While it is numerically fine, one could also argue that analysis biases are more consistent after the adjustment.  On the positive side, the *global drift* of bias during forecast seems much reduced.

The caption was missing stating that the verification is for the northern extra-tropics, this has been added. Similar results are seen for the southern extra-tropics and when verifying against sondes.  The paragraph has now been updated to read:

As previously noted much of the degradation in the forecast quality was due to a change in the bias of geopotential height forecasts in the troposphere. As well as the original results, Figure 2 shows the forecast bias for 500 hPa geopotential height of the experiments which apply a bias correction to the observations. The experiment which adjusts the observations by 0.05% approximately halves the negative bias in the short-range forecasts of this quantity. The experiment adjusting by 0.1% entirely eliminates the negative bias replacing with a slight positive bias, similar to the control NWP system. With increasing lead time, the forecast tends towards a positive 500 hPa geopotential height bias. The change of bias in the short-range forecast seems to be the main reason that the adjusted experiments perform better than the initial experiment — they are able to remove the large negative bias in the geopotential height forecasts. Figure 2 shows verification against ECMWF analyses in the northern extra-tropics. Similar results are seen in the southern extra-tropics and for verification against sondes.

- Page 15, section 2.5: please specify a "typical" (or reference) lead time of background forecasts.

We have added the following:

Since a 6h data assimilation window is used, the background forecast lead time is between 3 and 9h.

- Line 21: "... fit to the independent observations ..."
  What does "independent" refer to here? Does this express that these observations were not assimilated, and the set of observations in fig.12 is different from those in fig.11?

They are observations which are used in the assimilation (both the control and experiments), but are separate from GNSS-RO. However, we realise that the use of "independent" may have been misleading, so we have deleted it.

- Page 25, line 319: "... whereas the ECMWF system has little bias".
  Everybody may argue that this is true to a good extent, but some would rather say "... presumably has a smaller bias", or similar.

This remark is based on plots such as this from the ROM SAF monitoring:
https://rom-saf.eumetsat.int/monitoring/images/2025/2025-08-01/12_new/global_BA_GRAS-C_EUM_month.png
Hence, we're attempting to suggest a measurable difference between the two systems. We've updated the wording in an attempt to signify this:

The measured bending angle bias at high altitudes is different between the ECMWF and Met Office systems. The Met Office system has a positive bias at high altitudes, whereas the ECMWF system's observed bias is closer to zero.

- Line 320ff: "This may be partly due to the model ..."
  I find the description of the treatment of variables in the forward operator very confusing and not helpful. Could issues in the forward operator also explain the mean error patterns seen in fig.2/fig.15, or is there a relation?

The treatment in of variables in the forward operator affects the results that are seen in Figures 2 & 15. However, since we have not explored this in detail we don't wish to show any results. Rather, we have reworded the paragraph to hopefully be clearer:

The measured bending angle bias at high altitudes is different between the ECMWF and Met Office systems. The Met Office system has a positive bias at high altitudes, whereas the ECMWF system's observed bias is closer to zero. One feature of the Met Office system is that the forward operator doesn't ingest the forecast temperature directly. Instead it is provided with the air pressure and specific humidity, and the virtual temperature is derived from these. We conducted a very brief experiment which altered the operator to work directly from the temperature provided by the model indicated that this reduced the bias at high altitudes, bringing the statistics to be much closer to those of ECMWF (not shown). Therefore, applying this bias correction to the observations is unjustified since we are correcting the observations to look more like the model. However, applying a bias correction at high altitudes appears to be effective at improving the forecast quality, and therefore it is included in the experiments.

- Page 26, line 346ff: reference or personal communication?

Unfortunately, we don't have a reference for this – we are told that a paper is in preparation on the upgrade. We have added (Yan Liu, personal communication) to note this.

- Page 26, section 4.2: please specify the background lead time.

This has been added.

- Page 28, line 371ff: the text here and the caption of figs.22-25 are not consistent. The text refers to a change in RMSE, where the reader expects a decrease if the systems improves (similarly to fig.21 for the (O-B) statistics), while the figure captions refer to the "RMSE scorecard", where an increase denotes better results. Can the authors please resolve this?

As with the "overall" figure that is printed at the top of each scorecard, the values plotted in Figures 22 to 25 are positively-oriented (a reduction in the RMSE gives a positive overall figure). We accept that this is not clearly explained in the manuscript, so we have changed the presentation to be negatively-oriented, so that we are plotting the percentage reduction in RMSE. The figures have been changed and we feel this makes a lot more sense within the text.

- Page 30, line 388: "differences in locations of the verifying data points"
 It is not the locations alone but data density and spatial sampling that skews the verification against observations where each profile is both used and counted, while verification against analyses is less affected. I recommend to reformulate slightly.

We have reworded the beginning of this paragraph to account for this point. It now reads as follows:

There are many possible reasons which could lead to the differences noted above in the verification against sondes and ECMWF analyses. It seems likely that the

location and density of the verifying data points is an important factor.  Sondes in the northern extra-tropics are concentrated over land, and particularly over Europe...